# The Influence of Relative Reinforcing Value of Food, Sensitization, Energy Intake and Diet Quality on zBMI Change over Two Years in Adolescents: A Longitudinal Cohort Study

**DOI:** 10.3390/nu15092179

**Published:** 2023-05-03

**Authors:** Jennifer L. Temple, Tegan Mansouri, Ana Letícia Pereira Andrade, Amanda M. Ziegler

**Affiliations:** 1School of Public Health and Health Professions, University at Buffalo, Buffalo, NY 14214, USA; 2Department of Community Health and Health Behavior, School of Public Health and Health Professions, University at Buffalo, Buffalo, NY 14214, USA

**Keywords:** diet quality, reinforcing value of food, sensitization, energy intake, weight change, zBMI, adolescents

## Abstract

The relative reinforcing value (RRV) of food and sensitization are associated with zBMI and zBMI change over time, but the mechanisms underlying these relationships is unknown. The purpose of this study was to test the hypothesis that greater RRV and sensitization to HED food is associated with lower diet quality and greater energy intake at baseline and again at 24 months and that these relationships result in greater zBMI gain. The RRV of HED and LED food and dietary intake were measured at baseline and again after 24 months in a cohort of 202 boys and girls of 12–14 years old. The baseline RRV of HED food was associated with lower diet quality and lower energy intake at 24 months. zBMI gain was positively associated with the baseline energy intake but not baseline RRV of HED food or diet quality. However, diet quality moderated the relationship between baseline energy intake and zBMI change, with no difference in zBMI change as a function of energy intake when diet quality was high but significant and opposite relationships with energy intake when diet quality was low. This study suggests that high diet quality can reduce the negative impact of greater energy intake on zBMI change in adolescents.

## 1. Introduction

The relative reinforcing value (RRV) of food is a well-established, empirical index of motivation to obtain food. Greater RRV of highly palatable, high-energy-density (HED) foods is associated with higher energy intake, greater body weight and excess body weight gain over time [1]. While some cross-sectional associations between the RRV of food and energy intake have been established [1,2], little work has been conducted to explore the relationship between the RRV of food and diet quality or on the prospective relationships between these factors, and no work has explored how the RRV of lower energy density (LED) or healthier food relates to energy intake or diet quality. 

Understanding these relationships will help to determine how food RRV is involved in body weight regulation and to identify the optimal intervention targets to maximize healthy growth. The RRV of food is assessed using questionnaire or task-based methods that determine how hard an individual is willing to work to obtain a portion of a desired food [3]. Decades of research using this paradigm have shown that the RRV of food is a reliable [2] and valid [4] measure. Our previous work showed that the RRV of food can be altered by repeated snack food, with most adolescents and adults showing a decrease (satiation) in the RRV of food after repeated intake but with some showing an increase (sensitization) [5,6,7,8].

This sensitization phenotype is associated with increased risk of higher zBMI and excess zBMI gain over time [5,9,10], but the mechanisms for this relationship remain unclear. In addition, a previous analysis using the same cohort as this manuscript showed that the RRV of healthier food (LED) and sensitization to the repeated intake of LED food were not related to zBMI or zBMI change [9,10], suggesting that this phenotype is not protective against excess weight gain.

Changes in body weight are broadly related to energy balance with greater intake than expenditure leading to weight accumulation over time. Energy and nutrient intake can be assessed using analysis of 24 h dietary recalls. Although subject to recall bias, 24 h dietary recalls have been shown to be reliable [11] and valid [12] when gold-standard, multi-pass collection methods are used, in particular when multiple recalls across different days are combined [13]. These data can also be used to assess diet quality [14]. Lower diet quality scores are associated with food insecurity [15] and higher metabolic disease risk [16]. The relationship between diet quality score and weight status, however, is equivocal, with some studies showing no association [17,18] and others showing an inverse association [19,20]. Previous studies have not explored relationships between diet quality and the RRV of HED or LED food.

This study aimed to examine the relationships among RRV and sensitization to HED and LED food, diet quality, energy intake and zBMI/zBMI over a 2 y period in an observational cohort of 202 adolescents (conceptual model shown in Figure 1). In this longitudinal study, we independently tested relationships among baseline values and other measured outcomes after 24 months in the cohort. 

Through this work, we tested five a priori hypotheses: (1) Baseline RRV and sensitization to HED food are associated with greater energy intake and lower diet quality at 24 months. (2) Greater energy intake and lower diet quality at baseline are associated with greater zBMI gain over time. (3) Greater energy intake and lower diet quality at baseline are associated with greater RRV of HED food at 24 months. (4) Baseline RRV and sensitization to LED food are associated with lower energy intake and greater diet quality at 24 months. (5) Lower energy intake and higher diet quality at baseline are associated with greater RRV of LED food at 24 months. When taken together, these findings will help elucidate the mechanisms by which RRV of food, dietary intake and body weight are interconnected. 

## 2. Materials and Methods

General Study Description: This analysis was conducted as part of a two-year, prospective observational cohort study in 202 adolescents aimed at identifying behavioral factors associated with increased or decreased zBMI change over time. Data collection began in the summer of 2016 and was completed in the fall of 2021. The procedures have been described in detail previously [9,10]. The methods described here will be limited to the ones that are relevant for the current analyses. Briefly, participants visited the laboratory five times over a 5–6 week period. 

They completed a series of measures, including the relative reinforcing value (RRV) of high energy density (HED) and low energy density (LED) food, sensitization to repeated intake of HED and LED food, height and weight, 24 h dietary intake and demographics. After the baseline data collection visits, participants visited the laboratory for follow-up assessments at 6, 15 and 24 months. At all follow-up time points, height and weight were taken and a previous day, 24 h dietary recall was conducted. At the 24-month visit, the RRV of HED and LED food were also reassessed. All procedures described here were approved by the Social and Behavioral Science Institutional Review Board at the University at Buffalo. 

Study Participants: Potential participants were recruited from the greater Buffalo area using flyers in the community and in schools, e-mail outreach and direct mailings to homes. Parents of interested participants completed a screening survey, and eligibility was determined. Inclusion criteria were being within the age range, having a zBMI between −1.5 and 2.0 at baseline, moderate liking of at least one HED and LED food and willingness to consume those foods every day for two weeks, having a reasonable expectation of living in the Western New York area for the next two years, able to speak and read English (both participant and consenting parent or guardian). Exclusion criteria included medication use that could impact appetite or body weight (e.g., Adderall), medical conditions impacting appetite and allergies to the study foods. We used an upper limit of zBMI of 2 to exclude participants with obesity in accordance with WHO recommendations [21,22]. Study participants were 202 adolescents aged 12–14 years at baseline and 14–16 years at the 24-month follow-up timepoint. 

Anthropometrics: Height and weight were assessed in adolescents and parents/caregivers at baseline and again at the 6-, 15- and 24-month follow-up visits. Height and weight were assessed without shoes while wearing light weight clothing using a wall mounted, digital stadiometer (SECA; Hanover, MD) and a digital scale (SECA, Hanover, MD). Standardized zBMI values were then calculated using age- and sex-specific CDC growth charts [23].

Demographics: Parents completed a demographic questionnaire at baseline and follow-up visits at 6, 15 and 24 months. This questionnaire assessed parent and child race and ethnicity, marital status, household income, education of one or more parents and occupation.

Dietary Recall Procedures and Analysis: At all visits to the laboratory, participants were asked to recall all foods and beverages consumed the previous day. These data were collected using a multi-pass interview with a trained research assistant. The multipass method consists of five steps: (1) Quick list which is simply a list of all foods and beverages consumed, (2) a probe of forgotten foods, (3) time and occasions when food was consumed, (4) details about the amount of food, preparation methods and brands, and (5) a final review with all details and probing large gaps in time without consumption. Participants were also asked whether this was a typical eating day. 

If the participant reported that this was not a typical eating day, they were asked to provide a 24 h dietary recall of the most recent day that they felt was typical. If the participant was unsure about brands or preparation methods, the parent was asked to provide details, when possible. Dietary recall information was then entered into Nutritionist Pro (Version 7.4.0, 2019 Axxya Systems) by nutrition and dietetics graduate students to determine the energy and nutrient intake, and these data were reviewed by a registered dietitian. 

The information from three baseline dietary recalls was then averaged together, and that average was used as an estimate of the baseline energy and nutrient intake. We chose to use the average of three dietary recalls because prior research has shown that this improves accuracy [24]. The same procedures were used to assess energy intake and diet quality at 24 months, but we had only one dietary recall at this timepoint. 

Diet Quality: In order to estimate the quality of the diet, we used the USDA Food and Nutrient Database for Dietary Studies to code the number of servings of food in particular food groups [25]. Foods were first grouped into milk and dairy, protein foods, mixed dishes, grains, savory snacks, sweet snacks, fruits, vegetables, beverages, condiments and sauces and fats and oils, and we assessed how many servings per day of each group the participant reported consuming. We also assessed the total energy intake, intake of added sugar, saturated fat and sodium in grams and as a percentage of the total energy consumed. We then used the Dietary Guidelines for Americans [26] to determine if participants met or did not meet the guidelines based on their energy requirements and age. We assigned a score of ‘0’ for unmet and ‘1’ for met (Table 1). 

We focused on the food groups contained within these guidelines (fruit, vegetables, whole grain, protein, dairy, sodium, percentage of energy from added sugar and percentage of energy from saturated fat) for scoring. After scoring each section, diet quality was calculated based on the sum of scores of all food groups (scores range from 0 to 8). Each set of dietary recalls was analyzed and coded independently by two different nutrition or dietetics students. Once all scores were complete, the two spreadsheets were compared, and any discrepancies were resolved through discussion with the two coders and a registered dietitian.

Relative Reinforcing Value Task: The RRV of HED and LED food was assessed on visits 2–5 and visits 8 and 9 using a computer task that has been validated for use in children, adolescents and adults [9,10]. Briefly, participants were given the choice between responding on one computer for a portion of their preferred food (LED or HED) and the other computer for 2 min to engage in a highly liked sedentary activity (e.g., art activities, electronic games and puzzles). The LED (ED ≤ 1.0 kcal/g) food choices included: fruit cups, applesauce and low-fat yogurt. The HED (>4 kcal/g) food choices included: potato and corn chips, cookies and chocolate candies. 

Portions of food or preferred sedentary activity were earned on independent, progressive ratio schedules of reinforcement with 20 responses required for the first reinforcer and response requirements doubling thereafter (i.e., 20, 40, 80, 160, etc.). The highest schedule of reinforcement that was available required 5120 mouse clicks. Participants were instructed to work for their desired amount of food and time to engage in activity and that these points could be redeemed once responding was completed. They were also informed that they could not take food or activities out of the lab. The number of responses for each food on each schedule of reinforcement were plotted for each participant, and an area under the curve was generated. This was the measure used to calculate sensitization. 

Sensitization Paradigm: To assess sensitization to repeated intake of HED and LED food, we used the paradigm developed in our laboratory for use in adults [5,6,7,27] and adolescents [8,9,10]. Briefly, participants were given 14 portions of their preferred HED and LED food to take home and consume each day for two weeks between visits 2 and 3 and visits 4 and 5. The LED food portions were approximately 200 g/160 kcals, and the HED food portions were approximately 58 g/300 kcals. The order in which they received HED and LED food was randomized across participants. Participants were sent text reminders each day to consume their portion of food and they called, texted or completed an online survey reporting that they had consumed the food and when it was consumed. They were given no additional instructions on how or when to consume the food. All participants in this sample reported >70% compliance with the daily intake of both foods. Sensitization was calculated by subtracting the baseline area under the curve for the RRV of each food from the post-daily intake area under the curve for the RRV of each food. 

Sample Size Determination and Analytic Plan: The sample size for this study was based on our prior studies in adults [5,28] that examined the relationship between sensitization, BMI and weight change (effect size 0.19). We determined that, with an alpha of 0.05 and a power of 0.80, statistical significance could be achieved with a total of 180 participants. 

We first conducted cross-sectional analyses of baseline relationships and relationships at 24 months using Pearson Product–Moment Correlations. Specifically, we included pairwise comparisons of the energy intake, diet quality score, RRV of HED and LED food, sensitization of HED and LED food and zBMI at both time points. 

In order to test our hypotheses related to prospective relationships and to be able to account for missing values, we used multilevel regression modeling. To assess whether covariates were related to missingness, we examined potential differences between those with complete versus incomplete data. There were no significant differences between these groups for child and parent BMI, race, sensitization, pubertal development and food insecurity (all *p* > 0.05). We examined the scatterplots of zBMI data with each independent variable and observed that the data were linear. 

To test hypotheses related to baseline factors that predict diet, appetite and weight parameters at 24 months, we included sex, baseline zBMI, baseline energy intake, diet quality, RRV of HED and LED food and sensitization of HED food in all models. The dependent measures were the 24-month energy intake, diet quality, RRV of HED food and zBMI, and these were all tested in separate models. We repeated these models with RRV and sensitization of LED food. 

To test our hypothesis related to zBMI change over time, laboratory visits (level 1) were nested within individuals (level 2), and all models included visit number, sex and baseline zBMI as fixed effects. When our dependent measure was zBMI change, we included child zBMI at baseline and 6, 15 and 24 months. We used the number of months between the baseline visits and each follow-up visit as our marker of time (i.e., 6, 15 and 24). In order to test moderation in this analysis, we created interaction terms for diet quality and RRV of HED food, for diet quality and energy intake and for energy intake and RRV of HED food and included these in subsequent models. All data were analyzed using SPSS Version 27 (IBM Corp. Released 2020. IBM SPSS Statistics for Windows, Version 27.0. Armonk, NY, USA: IBM Corp), and data were considered significant if *p* < 0.05. 

## 3. Results

Participants: Descriptive characteristics of our sample at baseline are shown in Table 2. Briefly, our sample was evenly distributed between boys and girls (52% female), mostly non-Hispanic white (71%), higher income (52% reported a household income> $90,000/year) and mostly had parents who were college educated (68% completed college or an advanced degree). The 24-month timepoint includes 158 participants with complete data. The reason for the losses are as follows: voluntary withdrawal (n = 15), missed appointment/lost to follow-up (n = 24), began taking new medication that affected appetite (n = 3) and diagnosed with a new medical condition (n = 1).

Cross-sectional Correlations: There was a negative correlation between energy intake at baseline and baseline diet quality (β = −0.166; *p* < 0.001). There were positive correlations between baseline energy intake and the RRV of both HED (β = 0.177; *p* = 0.012) and LED (β = 0.139; *p* = 0.049) food but no relationships with HED or LED sensitization (both *p* > 0.05). There were no relationships between the RRV of food or sensitization with the 24-month energy intake. There were no correlations between the 24-month diet quality score and the RRV of HED or LED food at 24 months.

### Hypothesis Testing

**Hypothesis** **1.**Baseline RRV and sensitization of HED food is associated with greater energy intake and lower diet quality at 24 months. The baseline RRV of HED food was negatively associated with energy intake (β = −0.269; *p* = 0.016) and diet quality (β = −0.001; T(1, 580) = −2.925; *p* = 0.004) at 24 months. The baseline sensitization to HED food was also negatively associated with energy intake (β = −214.48; *p* = 0.009) and diet quality at 24 months (β = 0.472; *p* < 0.001). The baseline sensitization to HED food was positively associated with RRV of HED food (β = 0.989; *p* < 0.001) and zBMI (β = 0.169; *p* < 0.001) at 24 months. These data are shown in Table 3.

**Hypothesis** **2.**Greater energy intake and lower diet quality at baseline is associated with greater zBMI change over time. Greater energy intake at baseline was positively associated with zBMI gain at 24 months, even after controlling for baseline zBMI (β = 0.000; *p* = 0.046; Table 4). There was no relationship between the baseline diet quality or baseline RRV of HED food and zBMI gain over time (all *p* > 0.05).

When we examined the RRV of HED food and diet quality as potential moderators of the relationship between energy intake at baseline and zBMI change over time, we found that there was a significant interaction between diet quality and energy intake on zBMI change over time (β = −0.0008; *p* = 0.048) but no relationship with the RRV of HED food. Post hoc analyses were conducted using repeated measures ANOVA with “high” and “low” categories created for baseline energy intake and diet quality using median splits for each variable. These tests revealed that, when energy intake was low, diet quality did not greatly influence zBMI change; however, when energy intake was high, higher diet quality was associated with lower zBMI change, and low diet quality was associated with higher zBMI change (F(1, 101) = 4.66; *p* = 0.033; Figure 2).

**Hypothesis** **3.**Greater energy intake, RRV of HED food and sensitization to HED food are associated with greater RRV of HED food at 24 months. Greater energy intake at baseline was positively associated with the RRV of HED food at 24 months (β = 0.000; *p* = 0.018). Greater RRV of HED food (β = 0.002; *p* < 0.001) and sensitization to HED food (β = 1.19; *p* < 0.001) at baseline were also associated with greater RRV of HED food at 24 months.

**Hypothesis** **4.**Baseline RRV and sensitization to LED food are associated with lower energy intake and greater diet quality at 24 months. There was no association between the baseline RRV of LED food or baseline sensitization to LED food and energy intake or diet quality at 24 months (all *p* > 0.05).

**Hypothesis** **5.**Lower energy intake and higher diet quality at baseline is associated with greater RRV of LED food at 24 months. There was no relationship between the baseline diet quality or energy intake and RRV of LED food at 24 months (all *p* > 0.05).

## 4. Discussion

Our previous studies have shown that the RRV of HED food and sensitization of RRV of HED food are related to excess zBMI gain over time in adolescents, but the factors that result in weight change are poorly understood. We examined cross sectional and prospective relationships with RRV and sensitization to HED and LED food, energy intake and diet quality in a cohort of 202 adolescents who were followed for 24 months. In line with our first hypothesis, we found that the baseline RRV of HED food was negatively associated with diet quality and energy intake at 24 months. We also found that sensitization to HED food was positively associated with zBMI at 24 months but negatively associated with energy intake at 24 months. 

As hypothesized, zBMI gain over time was associated with greater baseline energy intake, however, contrary to our hypothesis, was not associated with the RRV of HED or LED food or diet quality. When we explored this relationship for potential moderators, we showed that the relationship between baseline energy intake and zBMI gain was moderated by diet quality but not by the RRV of food. Finally, the baseline RRV of LED and sensitization to LED food were not related to the energy intake, diet quality or zBMI at 24 months.

Weight gain over time is the result of a positive energy balance that is typically driven by excess energy intake. In the current study, we showed that a greater energy intake at baseline was associated with greater zBMI gain over time. We were interested in identifying moderating factors for this relationship and, therefore, examined the RRV of HED food and diet quality as potential moderators. We found that diet quality, but not RRV of HED food, interacted with energy intake to influence zBMI gain over time. Specifically, when diet quality was low, higher energy intake resulted in zBMI gain, while lower energy intake resulted in zBMI loss over 2 years. However, when diet quality was high, zBMI stayed steady over the two-year period when energy intake was low and decreased when energy intake was high. 

This finding suggests that having access to a high-quality diet can reduce the risk of excess zBMI change, even in the presence of greater energy intake. These findings are consistent with those previously reported, including a study by Hu and colleagues that showed that adolescents who consumed a high-quality diet gained less weight over time as they transitioned to adulthood [29]. Another study by Kevin Hall and colleagues in adults showed that energy intake and weight gain were higher when participants consumed a lower quality diet [30]. When taken together, these data suggest that improving the quality of the diet might be an important step in reducing excess weight gain during adolescence.

We also examined whether the RRV of HED food acted as a moderator of the relationship between energy intake and zBMI change, but we found that it did not. Prior research has shown that the RRV of HED food is cross-sectionally and prospectively associated with weight status in both children [31,32,33] and adults [34,35]. Further, energy intake and the RRV of HED food have been shown to independently relate to weight status [33], but this is the first study to examine whether the RRV of food moderates the relationship between energy intake and weight change. 

Our data suggest that the RRV of food acts parallel to energy intake as an independent factor influencing weight change, but that they do not work together. The baseline RRV of food was also not associated with increased energy intake at 24 months, but it was negatively associated with diet quality at baseline and at 24 months, suggesting that a greater motivation to consume HED snack foods might lead to a reduction in diet quality.

While previous studies have shown that higher intake of fruits and vegetables increases weight loss [36], no previous studies have examined the RRV of LED food and weight status. In this study, we sought to address this gap by examining the relationships among the RRV of LED food and diet quality, energy intake and weight change. We found no relationships among RRV of LED food and diet quality or energy intake at baseline or 24 months. There was also no relationship between the RRV of LED food and zBMI change. 

One explanation for the lack of a relationship between the RRV of LED food and diet quality and weight is that a high RRV of LED food is not mutually exclusive from a high RRV for HED food and that the RRV of HED food might play a more important role in driving energy intake, diet quality and weight change. In other words, people with a high RRV of LED food might also have a high RRV of HED food, and RRV of HED food is a larger driver of energy intake, diet quality and weight. These data suggest that prevention and intervention strategies that focus on increasing the intake of LED foods alone will not facilitate weight management or improve diet quality.

This study had several strengths, including a large sample of adolescent participants, a rigorous, laboratory-based experimental design with the provision of snack food and objective measures of anthropometrics and behaviors and an in-depth analysis of diet quality based on repeated 24 h dietary recalls. 

This study also had certain limitations. First, the sample was largely white and had a higher income and education level than the general population. This makes it difficult to generalize our findings to low-income and minority adolescents. One important difference in this case could be related to food access. This study was conducted in food-secure adolescents who had easy access to a wide variety of HED and LED foods. The outcomes may differ in a food-insecure population. Future studies should examine these relationships further. 

Second, the attrition from baseline to 24 months was approximately 30%, which was higher than we anticipated and may have limited our power to detect differences. Third, because of limitations in our nutrient analysis software, we were unable to calculate the HEI score for our participants. This makes it difficult to compare our assessment of diet quality with other published work. Finally, while the diet quality and energy intake data were based on self-reported, interviewer-led 24 h dietary recalls, it is possible that adolescents underreported and/or mis-reported their intake.

## 5. Conclusions

When taken together, this study adds to our understanding of factors that influence weight change in adolescents. We demonstrated that low diet quality interacts with energy intake to influence zBMI gain, but high diet quality can buffer adolescents from increased zBMI. This suggests that focusing on improving diet quality in youth as opposed to limiting energy intake may be an intervention strategy that can reduce excess weight gain while limiting the negative consequences of energy restriction in this age group. 

Future studies will need to investigate whether changing diet quality among adolescents is feasible, in particular for lower income populations and adolescents with increased eating autonomy. Further research should also address how reducing stigma around weight gain as strictly related to HED intake and poor diet quality could generate promising findings, therefore, assessing not only the individual level but also environmental and social factors.

## Figures and Tables

**Figure 1 nutrients-15-02179-f001:**
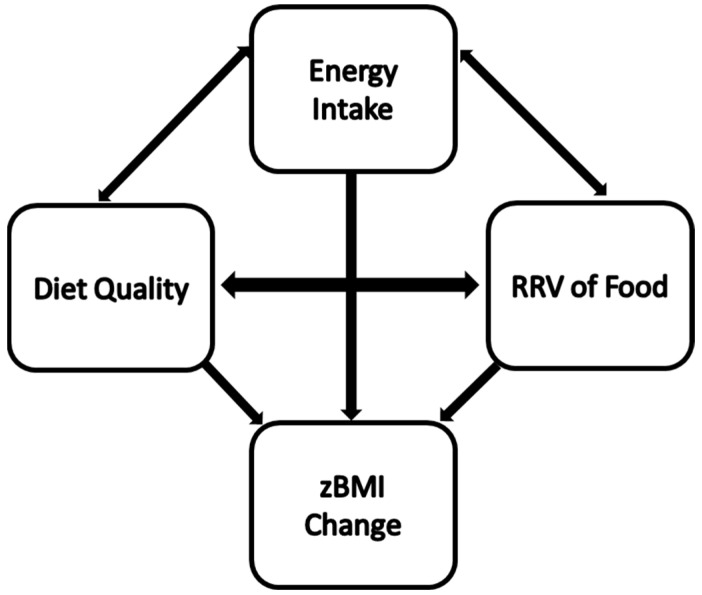
Conceptual model of the relationships between energy intake, diet quality, RRV of food, and zBMI change over time.

**Figure 2 nutrients-15-02179-f002:**
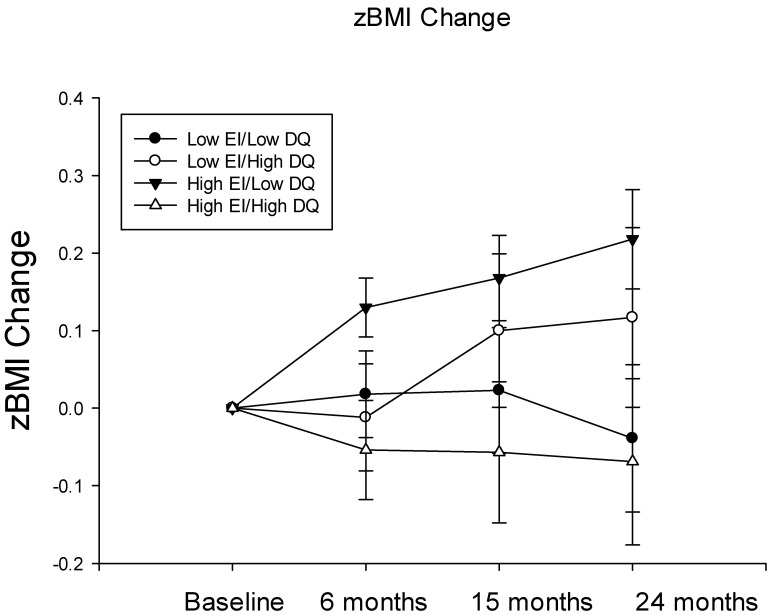
The mean ± SEM zBMI change from baseline at 6, 15 and 24 months in participants who had low (circles) or high (triangles) energy intake and low (black) or high (white) diet quality at baseline.

**Table 1 nutrients-15-02179-t001:** The Dietary Guidelines for Americans criteria.

Dietary Intake Variable	Guideline (Number of Servings Based on Estimated Energy Needs; kcals)	Mean (SD)	% Meeting Guideline
DIETARY COMPONENTS TO EAT			
Fruits (cups/day)	1400–1800-1.5 2000–2600-2	1.5 (1.9)	37%
Vegetables (cups/day)	1400-1.5 1600-2 1800–2000-2.5 2200–2400-3 ≥2600-3.5	1.4 (1.3)	24%
Whole grain (oz eq/day)	1400-2.5 1600–2000-3 2200-3.5 2400-4 2600-4.5	0.5 (0.8)	4%
Dairy (cup eq/day)	3 servings	1.4 (1.2)	11%
Protein (ounce eq/day)	1400-4 1600–1800-5 2000-5.5 2200-6 2400–2600-6.5	8.9 (5.8)	78%
DIETARY COMPONENTS TO LIMIT			
Added Sugar	<10% of energy per day	15.1 (11.3)	34%
Saturated Fat	<10% of energy per day	11.8 (2.8)	28%
Sodium	<2300 mg/day	2523 (1063)	47%

**Table 2 nutrients-15-02179-t002:** Participant characteristics. Participant data are shown with the number of participants and percentages for sex, race/ethnicity, household income and parental education. We also show descriptive data for baseline and follow-up time periods for age, zBMI, pubertal development score, diet quality and energy intake.

	n = 202	
	n	%
Sex Male Female	97 105	48 52
Ethnicity Hispanic or Latino Non-Hispanic or Latino	15 186	7 93
Race American Indian/Alaska Native Asian/Pacific Islander Black/African American White or Caucasian Other/mixed race	1 4 19 159 17	1 2 9 79 8
Household Income <$9999 $10,000–$49,999 $50,000–$69,999 $70,000–$89,999 $90,000–$109,999 $110,000–$139,999 >$140,000	1 33 31 31 29 36 40	0.5 16 15 15 14 18 20
Parental Education Completed high school Some college/completed vocational training Complete college/university Completed graduate degree	9 28 71 67	4 14 35 33
	Mean	SEM
Baseline Descriptives		
Age (years)	13.3	0.06
zBMI	0.396	0.07
Pubertal Development Score	2.6	0.05
Diet Quality Score	3.4	0.10
Energy Intake (kcals)	1640.2	34.5
Two-Year Follow-Up Descriptives		
Age (years)	15.25	0.064
zBMI	0.443	0.007
Pubertal Development Score	3.23	0.04
Diet Quality Score	1.9	0.09
Energy Intake (kcals)	1815.7	65.6

**Table 3 nutrients-15-02179-t003:** Regression models predicting the 24-month energy intake, diet quality, RRV of HED food and zBMI.

	β	SE	T	F	*p*
Model 1—Energy Intake at 24 Months					
Sex	−290.6	63.44	−4.58	20.98	<0.001
Baseline zBMI	−84.21	32.23	−2.61	6.79	0.009
V1 Energy Intake	0.39	0.045	8.76	76.68	<0.001
V1 Diet Quality	−17.46	24.54	−0.71	0.506	0.477
V1 RRV of HED Food	−0.346	0.112	−3.08	9.51	0.002
V1 Sensitization of HED Food	−168.07	84.59	−1.99	3.95	0.047
Model 2—Diet Quality at 24 Months					
Sex	0.168	0.103	−1.31	2.66	0.1.04
Baseline zBMI	−0.069	0.053	−1.31	1.71	0.191
V1 Energy Intake	0.000	0.000	1.76	3.082	0.080
V1 Diet Quality	0.007	0.040	0.185	0.034	0.853
V1 RRV of HED Food	−0.001	0.000	−4.91	24.12	<0.001
V1 Sensitization of HED Food	0.472	0.140	3.37	11.34	<0.001
Model 3—RRV of HED Food at 24 Months					
Sex	−0.606	0.118	−5.14	26.42	<0.001
Baseline zBMI	−0.066	0.06	−1.1	1.22	0.27
V1 Energy Intake	0.000	0.000	2.23	4.99	0.026
V1 Diet Quality	−0.019	0.046	−0.403	0.162	0.687
V1 RRV of HED Food	0.002	0.000	9.56	91.35	<0.001
V1 Sensitization of HED Food	1.19	0.165	7.24	52.46	<0.001
Model 4—zBMI at 24 Months					
Sex	0.12	0.04	2.99	8.99	0.003
Baseline zBMI	0.817	0.02	40.52	1641.98	<0.001
V1 Energy Intake	−0.0005	0.0005	0.123	0.015	0.902
V1 Diet Quality	−0.041	0.016	−2.66	7.056	0.008
V1 RRV of HED Food	0.0003	0.0006	0.323	0.104	0.747
V1 Sensitization of HED Food	0.125	0.056	2.23	4.98	0.026

**Table 4 nutrients-15-02179-t004:** Regression model examining the factors that relate to zBMI change over time. Baseline factors of sex, zBMI, energy intake, diet quality, RRV of HED food as well as interactions among diet quality, energy intake and RRV of HED food (interactions indicated with *) in order to examine moderators. Significant predictors of zBMI change over time are highlighted in gray.

	β	SE	T	F	*p*
zBMI Change Over Time					
Sex	0.097	0.035	2.786	7.759	0.006
Baseline zBMI	0.893	0.025	36.19	1310.15	<0.001
V1 Energy Intake	0.000	0.000	2.081	4.33	0.039
V1 Diet Quality	0.084	0.068	1.23	1.51	0.22
V1 RRV of HED Food	0.000	0.000	−1.66	2.75	0.099
Diet Quality ∗ Energy Intake	−0.0008	0.0003	−1.99	3.99	0.048
Diet Quality ∗ RRV of HED Food	0.00003	0.0008	1.675	2.81	0.096
Energy Intake ∗ RRV of HED Food	−0.0001	0.00002	−0.680	0.604	0.497

## Data Availability

The data and statistical analysis syntax will be made available upon request.

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
