# Peer review of "The Influence of Relative Reinforcing Value of Food, Sensitization, Energy Intake and Diet Quality on zBMI Change over Two Years in Adolescents: A Longitudinal Cohort Study"

_nutrients, 2023, doi:10.3390/nu15092179_

Round 1
Reviewer 1 Report
The search for the relationship between the quality and composition of the diet and the energy of the diet and changes in body weight in adolescents is intentional and justified.
For a better characterization of the conducted study and discussion of its results, it would be worth describing the method of selecting the study group (despite references to earlier works) and providing the energy value of the diet and its changes as well as the value of zBMI and its changes over two years. An additional factor that could have influenced the observed relationships was the exclusion of adolescents with obesity. Similarly, the past 24-hour dietary recall could be replaced by recording intake for three non-consecutive days.
Detailed comments:
· Why was the diet baseline energy intake taken as the average of three 24-hour dietary recalls obtained, according to the authors description, at different visits?
· What was the consumption of foods classified as LED and HED with the diet, in addition to the products that the teenagers were given? Similarly, it is worth specifying the changes in zBMI and HEI values over 2 years.
· In Tables 2, 3 and Figure 2, it is worth adding explanations of the abbreviations used.
· In the chapter on the second hypothesis, does the term "diet quality" refer to the value of the Healthy Eating Index?
Author Response
- “it would be worth describing the method of selecting the study group (despite references to earlier works)” We have added additional detail about how we recruited and selected participants to the methods. “Potential participants were recruited from the greater Buffalo area using flyers in the community and in schools, e-mail outreach, and direct mailings to homes. Parents o interested participants completed a screening survey and eligibility was determined. Inclusion criteria were being within the age range, having a zBMI between -1.5 – 2.0 at baseline, moderate liking of at least one HED and LED food and willingness to consume those foods every day for two weeks, having a reasonable expectation of living in the Western New York area for the next two years, able to speak and read English (both participant and consenting parent or guardian). Exclusion criteria included medication use that could impact appetite or body weight (e.g. Adderall), medical condition impacting appetite, and allergies to the study foods. We used an upper limit of zBMI of 2 to exclude participants with obesity in accordance with WHO recommendations. [21, 22] Study participants were 202 adolescents aged 12 – 14 at baseline and 14 – 16 at the 24 month follow up timepoint.”
- “providing the energy value of the diet and its changes as well as the value of zBMI and its changes over two years.” We have added more information to the participant characteristics table, including zBMI at baseline and 24 months, energy intake at these timepoints, and diet quality scores.
- “Why was the diet baseline energy intake taken as the average of three 24-hour dietary recalls obtained, according to the authors description, at different visits?” We have added the following to the manuscript: “The information from three baseline dietary recalls was then averaged together and that average was used as an estimate of baseline energy intake and diet quality. We chose to use the average of three dietary recalls because prior research has shown that this improves accuracy (REF). The same procedures were used to assess energy intake and diet quality at 24 months, but we only had one dietary recall at this timepoint. “
- What was the consumption of foods classified as LED and HED with the diet, in addition to the products that the teenagers were given? We did not analyze the dietary recalls based on energy density. We chose to use the Dietary Guidelines for Americans to determine if the recommendations were met or unmet. We have provided more details about this in the revised version.
- Similarly, it is worth specifying the changes in zBMI and HEI values over 2 years. We have added this information to the Table 2.
- In Tables 2, 3 and Figure 2, it is worth adding explanations of the abbreviations used. We have added descriptions of the abbreviations.
- In the chapter on the second hypothesis, does the term "diet quality" refer to the value of the Healthy Eating Index? In response to Reviewer 2, we have added much more detail about our assessment of diet quality. We have removed references to the HEI and have instead referred to diet quality throughout.
Reviewer 2 Report
Results of this study may contribute to understanding of factors associated with body weight in adolescents and potentially inform interventions designed to promote healthy weight. Overall, the paper is well-written and results are clearly presented and discussed.
However, a major concern is that the methods used to determine the Healthy Eating Index appear to be incorrect. In Line 133, it says that USDA Healthy Eating Index guidelines were used, but there is no reference. And then the USDA Choose My Plate guidelines are referred to in Line 138. This should be referenced. But secondly, these are different than the HEI scoring guidelines. Using the appropriate method may or may not change some of the results and conclusions. All of the information needed for determining the correct method to use and how to calculate the HEI can be found at https://epi.grants.cancer.gov/hei/.
There needs to be much more information provided in the methods about how diet quality was estimated, regardless of whether it was the appropriate HEI method or the authors' approach. The current approach for estimating diet quality as described can be retained, but it cannot be called the Healthy Eating Index. And authors should indicate why they believe their approach is a reliable indicator of diet quality as well as discuss relevant limitations in the discussion. Authors indicate that intakes were compared against recommended intakes of fruit, vegetables, whole grain, refined grain, protein, and dairy. How were intakes of sweet foods and sweetened beverages, snack foods, fats and oils considered? Were recommendations for food group intakes based on individual energy requirement? How were mixed dishes handled?
Author Response
However, a major concern is that the methods used to determine the Healthy Eating Index appear to be incorrect. In Line 133, it says that USDA Healthy Eating Index guidelines were used, but there is no reference. And then the USDA Choose My Plate guidelines are referred to in Line 138. This should be referenced. But secondly, these are different than the HEI scoring guidelines. Using the appropriate method may or may not change some of the results and conclusions. All of the information needed for determining the correct method to use and how to calculate the HEI can be found at https://epi.grants.cancer.gov/hei/.
There needs to be much more information provided in the methods about how diet quality was estimated, regardless of whether it was the appropriate HEI method or the authors' approach. The current approach for estimating diet quality as described can be retained, but it cannot be called the Healthy Eating Index. And authors should indicate why they believe their approach is a reliable indicator of diet quality as well as discuss relevant limitations in the discussion. Authors indicate that intakes were compared against recommended intakes of fruit, vegetables, whole grain, refined grain, protein, and dairy. How were intakes of sweet foods and sweetened beverages, snack foods, fats and oils considered? Were recommendations for food group intakes based on individual energy requirement? How were mixed dishes handled?
- We thank the reviewer for these comments and have provided a lot more detail and clarity around the methods that were used. As an estimate of diet quality, we first used the What We Eat in America/Healthy Eating Index guidelines to categorize foods and determine the number of servings of each food group eaten. Unfortunately, the diet analysis software that we used (Nutritionist Pro) does not provide some of the data needed to calculate the Healthy Eating Index, so we instead used the Dietary Guidelines for Americans and compared the intake of each of the major groups recommended by the DGA (fruits, vegetables, dairy, protein, and whole grain) with the intake of our participants. We created a coding scheme that reflected not eating at all (-1), eating some, but below the recommended amounts (0), or meeting/exceeding the recommendations (1). After reading these reviews, we returned to the data and recalculated our diet quality scores to include a consideration of meeting recommendations for nutrients to limit (sodium, saturated fat, and added sugar). We recoded all food groups as met (1) or unmet (0) in terms of the recommendations based on age and energy needs. We reanalyzed all data. While some numbers shifted slightly with this new coding scheme, the primary relationships remain. We have added more information about these calculations to the methods and have added a table explaining these calculations.
“Diet Quality: In order to estimate the quality to the diet, we used the What We Eat In America guidelines to categorize foods into different food groups (REF)> Foods were first grouped into : milk and dairy, protein foods, mixed dishes, grains, savory snacks, sweet snacks, fruits, vegetables, beverages, condiments and sauces, and fats and oils and assessing how many servings per day of each group the participant reported consuming. We also assessed total energy intake, intake of added sugar, saturated fat, and sodium in grams and in percentage of total energy consumed. We then used the USDA Choose My Plate guidelines to determine if participants met or did not meet the guidelines, based on their energy requirements and age. We assigned a score of ‘0’ for unmet and ‘1’ for met (Table 1). We focused on the food groups contained within these guidelines (fruit, vegetables, whole grain, protein, dairy, sodium, percent of energy from added sugar and percent of energy from saturated fat. After scoring each section, diet quality was calculated based on the sum of scores of all food groups (scores range from 0 – 7). Each set of dietary recalls was analyzed and coded independently by two different nutrition or dietetics students. Once all scores were complete, the two spreadsheets were compared and any discrepancies were resolved through discussion with the two coders and a registered dietitian.”